# Prevalence and Associated Factors of Erosive Tooth Wear among Preschool Children—A Systematic Review and Meta-Analysis

**DOI:** 10.3390/healthcare10030491

**Published:** 2022-03-07

**Authors:** Kimberley Yip, Phoebe Pui Ying Lam, Cynthia Kar Yung Yiu

**Affiliations:** Paediatric Dentistry, Faculty of Dentistry, The University of Hong Kong, Hong Kong; kimberley.yip@gmail.com (K.Y.); phoebe17@connect.hku.hk (P.P.Y.L.)

**Keywords:** tooth erosion, child, preschool, prevalence, risk factors, systematic review

## Abstract

The prevalence of dental erosion among preschool children and its associated factors range widely between studies. The aims of this review are to evaluate the literature and to determine the prevalence and associated factors of dental erosion among children below 7 years old. An electronic search was undertaken to identify observational studies evaluating the prevalence of dental erosion and its associated factors in children below 7 years old. Dual independent screening, data extraction, risk of bias assessment, meta-analysis, meta-regression, and evaluation of quality of evidence were performed. Twenty-two papers were included. The overall estimated prevalence of dental erosion in children was 39.64% (95% CI: 27.62, 51.65; I^2^ = 99.9%), with very low certainty of evidence. There was also low-quality evidence suggesting that the likelihood of (1) boys having dental erosion was significantly higher than girls (*p* < 0.001) and (2) children with digestive disorders having dental erosion was significantly higher than those without such digestive disorders (*p* = 0.002). Qualitative synthesis identified that more frequent intake of fruit juices and soft drinks correlated with erosive tooth wear. Dental erosion is prevalent among over one-third of preschool children. Digestive disorders and dietary factors are the main potential contributing factors.

## 1. Introduction

Dental erosion refers to the chemical loss of mineralized tooth substance caused by the exposure to acids not derived from oral bacteria [1]. Dental erosion can cause dentine hypersensitivity; poor aesthetics; or in severe cases, near or frank pulp exposures requiring root canal treatment or extraction [2]. As enamel of the primary teeth is softer than that of permanent teeth [3], erosive tooth wear in primary teeth may occur faster and possibly result in pulp exposure in some cases [4]. As erosive tooth wear has serious long-term implications, it is important to establish the prevalence of erosive tooth wear, and its associated and aetiological factors.

Erosive tooth wear is understood as a significant problem to many adults. Almost 80% of Swedish and up to 97.9% of Chilean adults have signs of erosive tooth wear [4,5]. The prevalence of erosive tooth wear in children is, however, inconsistent. The prevalence ranges from 5.7% to 78% [6,7], depending on the study location, age, methodology, as well as definition and criteria of “erosive tooth wear”. There have been multiple reviews on the risk factors of erosive tooth wear in adults [8,9] and adolescents [10]. Risk factors associated with erosive tooth wear in these age groups include gastroesophageal reflux disease (GERD) [8,9], eating disorders associated with vomiting [9], vegetarian diets [11], and frequent consumption of soft drinks and acidic drinks, particularly at bedtime [10]. 

However, there has not been a recent review on the associated factors of erosive tooth wear in children. Individual studies have conflicting conclusions regarding the risk factors of erosive tooth wear in children. One study found that dietary factors, oral hygiene behaviour, and systemic diseases were not associated with erosion in children [12], while another study [13] found that the frequent consumption of certain foods, including fruit juice and citrus fruits, was significantly associated with erosive tooth wear. Similar to adults, children and adolescents with certain medical conditions are also more likely to have erosive tooth wear. For instance, children with gastroesophageal reflux disease (GERD) had an increased risk of erosive tooth wear compared with children without GERD [14]. The level of parental education may also be a factor associated with erosive tooth wear that is unique to children [7,15]. As previous studies have raised a multitude of possible risk factors of erosive tooth wear in children, the aim of this review is to systematically evaluate the literature and to determine the prevalence and associated factors of erosive tooth wear among children.

## 2. Materials and Methods

### 2.1. Study Design

The protocol for this systematic review was written and registered on PROSPERO (registration number: CRD42020186982). The review was conducted in accordance with the Preferred Reporting Items for Systematic Reviews and Meta-Analyses (PRISMA) Guidelines (Appendix A).

### 2.2. Eligibility Criteria

The inclusion criteria are listed here following the PECO(S) statement:
Population (P): individuals with primary dentition up to age 6;Exposure (E) and Comparison (C) factors: gender, dental habits (such as frequency of brushing of teeth), dietary habits (such as frequency of consumption of fruit juice, soft drinks, and fruits), medical conditions (such as GERD and vomiting), and parental education level;Outcome (O): prevalence of erosive tooth wear, measured by different erosion indices such as Basic Erosive Wear Examination (BEWE), Tooth Wear Index (TWI), O’Sullivan’s Index, and Erosion Partial Recording System (EPRS); andStudies (S): observational studies, such as cohort, case–control, and cross-sectional studies on the prevalence and risk factors of erosive tooth wear in children up to 7 years of age with full texts written in English.

The exclusion criteria included the following:
Irrelevant studies;Studies on other types of tooth wear that were not erosive tooth wear;Studies on diagnosis and management of erosive tooth wear only;Studies reporting prevalence without conducting a clinical examination;Studies including individuals older than 7 years old or those with special needs;Case reports, clinical trials, laboratory studies, conference papers, and reviews; andStudies with non-extractable data, studies not in English, and duplicated studies.

### 2.3. Search Strategy

A systematic electronic search was undertaken. Four electronic databases (Ovid MEDLINE, Ovid Embase, Web of Science, and Scopus) were searched from inception to July 2020. The subject index terms and MeSH terms used for the search were as follows: tooth erosion, dental erosion, acid erosion, enamel erosion, erosive tooth wear, dental erosive wear, prevalence, risk factors, risk assessment, risk factors assessment, risk factor function, indicating factors, erosive factors, causative factors, boys, girls, child, children, childhood, pediatric, adolescent, teenage, and youth (Appendix A). The search was limited to articles in English and articles with English translations. Manual searches of articles were conducted by scrutinizing the reference lists of relevant reviews.

### 2.4. Selection of Studies

After the initial search, duplicate papers were removed. Two independent reviewers (first and second authors) screened for relevant reports by reading the titles and abstracts. Next, the two reviewers read the full texts of the reports and independently selected eligible articles based on the inclusion and exclusion criteria that were previously discussed. Kappa (κ) statistics was used to evaluate the degree of agreement between both reviewers. Any disagreements were discussed between both reviewers until a consensus was reached.

### 2.5. Data Extraction

Both reviewers systematically extracted relevant data from the selected articles and organized them into tables. The data that were extracted included details of the participants (country of origin, gender, age, medical conditions such as GERD, vomiting, and parental education level), and exposure (dental habits, and dietary habits such as frequency of fruit juices and soft drinks).

### 2.6. Assessment of Risk of Bias

The Joanna Briggs Institute (JBI) critical appraisal checklist for studies reporting prevalence data was used to assess the quality and risk of bias of included studies [16,17]. The methodology of each study was evaluated by answering each question in the checklist with “yes”, “no”, “unclear”, or “not applicable”. Studies with a score of 70% and above were considered to have a low risk of bias, while studies with a score of 50–69% and below 50% were considered to have moderate and high risk of bias, respectively. Both reviewers independently completed the JBI critical appraisal checklist for each study. Any disagreements were resolved by discussion between reviewers.

### 2.7. Data Synthesis

Using random-effects models, meta-analyses (Stata version 13.1, StataCorp, College Station, TX, USA) was performed for prevalence of erosive tooth wear and for dichotomous outcomes with only two groups. This included gender, presence of digestive disorders, birthplace (local or foreigner), presence of caries, parental education (primary education and below vs. above, and secondary education and below vs. above), and toothbrushing (once or less than once a day vs. more than once a day). All studies were included in the meta-analysis. Sensitivity analyses were conducted, and studies with high risk of bias were excluded to determine whether the quality of included studies had any effect on the results. For outcomes with more than two groups, such as age, indices, sample size, human development index (HDI), and year of recruitment, meta-regression was used to examine these possible risk factors with random-effects models. The results were presented narratively when outcome measures were too heterogeneous.

### 2.8. Assessment of Heterogeneity

Heterogeneity of the outcome results were assessed by following the Cochrane Handbook for Systematic Reviews of Intervention [18,19]. I^2^ statistics and chi-square test were used to calculate the amount of heterogeneity and level of significance (*p* < 0.05). Heterogeneity was considered as significant if I^2^ > 50% or χ^2^
*p* < 0.05.

### 2.9. Assessment of Publication Bias

A Funnel plot and Egger’s test were used to test for any potential publication bias when ten or more studies contributed to the outcome [20,21].

### 2.10. Assessment of Quality of Evidence

The Grading of Recommendations Assessment Development and Evaluation (GRADE) approach [22] was used to independently evaluate the quality of evidence for each outcome. As only observational studies were included, the quality of evidence was considered low from the start. The quality of evidence was downgraded if there was evidence of risk of bias, inconsistency, indirectness, imprecision, or publication bias. The quality of evidence was upgraded if there was a large magnitude of effect (i.e., strong association), a presence of a dose–response gradient, and also plausible confounding factors [22].

## 3. Results

### 3.1. Study Selection

An initial literature search yielded 1523 articles. After duplicate articles were removed, 745 articles remained. The title and abstract of these records were screened independently by both reviewers. The full text of the remaining 152 articles were assessed for eligibility. Another 130 articles were excluded due to reasons such as subjects being above 6 years old (*n* = 90), full texts being unavailable (*n* = 11), and full texts being in languages other than English (*n* = 8) (Appendix A). A total of 22 studies [7,13,15,23,24,25,26,27,28,29,30,31,32,33,34,35,36,37,38,39,40,41] met the inclusion criteria and were included for qualitative and quantitative analyses (κ = 0.937). The screening and study selection process are presented in the PRISMA flowchart (Figure 1).

### 3.2. Study Characteristics

The study characteristics are reported in Table 1. All studies were observational studies. A total of 17,300 subjects from 15 countries across Asia, Europe, and North and South America were included in this review. A majority of the studies (72.72%) recruited subjects from kindergartens and schools [7,13,15,24,25,27,28,29,30,31,33,34,35,39,40,41], while the remaining subjects were recruited from clinics (13.64%) [23,32,38], a compulsory national vaccination day (9.09%) [36,37], and a home for the disadvantaged (4.54%) [26]. Factors associated with erosion that were evaluated in the studies varied, but they mainly included diet, oral hygiene, socioeconomic status, parental education level, age, gender, and caries. Erosion was measured by Tooth Wear Index (TWI), O’Sullivan Index, Basic Erosive Wear Examination (BEWE), Erosion Partial Recording System (EPRS), and their modifications. Over two-thirds (68.18%) of the included studies used TWI and TWI-modified index [7,13,24,25,26,27,29,30,31,32,35,36,37,39], while the remaining papers used EPRS [23], O’Sullivan Index [38,40], BEWE [15,28,33,34,41], and their respective modified indices (Appendix A).

### 3.3. Risk of Bias

Using the JBI critical appraisal checklist for prevalence studies, the risk of bias of each included study was evaluated and reported in Table 2. Out of the twenty-two studies, thirteen (59.09%) had low risk of bias [13,15,25,26,27,28,30,33,34,36,37,40,41], eight (36.36%) had moderate risk of bias [7,23,24,29,31,32,38,39], and one (4.55%) had a high risk of bias [35].

Almost all studies used an appropriate sample frame to address the target population (Q1), except for two studies [23,38]. One study [23] had a target population that consisted of “Yemeni children and adolescents”, but the sampling frame only included children and adolescents who visited the university’s dental clinic. Similarly, another study [38] targeted “Japanese children” but had a sample frame that only consisted of patients who attended the outpatient paediatric dental clinic in a hospital.

Most studies sampled study participants in an appropriate way (Q2). However, there were three studies [25,31,35] that did not clearly state their method of sampling. Convenience sampling was used in one study [29], and therefore, this study was deemed as not employing an appropriate way of study participant recruitment. Regarding sample size (Q3), seven [7,29,31,32,35,38,39] out of the twenty-two studies did not show evidence that the authors conducted a sample size calculation to determine an adequate sample size. Therefore, it was deemed “unclear” whether the sample size was adequate.

Regarding subject and setting description (Q4), nine studies [23,24,25,28,29,32,35,37,41] did not describe the study subjects and setting in detail. In ten studies [7,13,23,30,32,33,34,37,38,39], it was unclear if a data analysis was conducted with sufficient coverage of the identified sample (Q5). There was insufficient coverage of the identified sample in two studies [24,35]. As such, there could have been coverage bias in the above twelve studies.

All studies used valid erosive tooth wear indices to identify and measure erosive tooth wear (Q6), and appropriate statistical analysis to evaluate the findings (Q8). Erosion was measured in a standard and reliable way for all participants (Q7) in most studies, except for one study (Raza & Hashim, 2012), which did not mention who collected the data, and whether the examiners had been trained or calibrated. A majority of the studies [7,23,24,27,28,30,31,32,34,35,38,39,40] did not mention whether the response rate was adequate (Q9) and if the low response rate was managed appropriately. The completed JBI appraisal for each study is detailed in Table 2.

### 3.4. Prevalence of Erosive Tooth Wear

All twenty-two studies contributed to the meta-analysis on the prevalence of dental caries [7,13,15,23,24,25,26,27,28,29,30,31,32,33,34,35,36,37,38,39,40,41]. The estimated combined prevalence of erosive tooth wear in children below 7 years old is 39.64% (95% CI: 27.62, 51.65), as seen from Figure 2. There was significant heterogeneity across the studies, with I^2^ of 99.9%, and *p* < 0.001. As more than 10 studies contributed to this overall estimate, Begg’s and Egger’s tests were used to check for publication bias. Begg’s and Egger’s tests showed no significant bias (*p* = 0.866 (continuity corrected), *p* = 0.444, respectively). Following a GRADE recommendation, the body of evidence was downgraded to a very low quality of evidence due to the data being observational and having substantial inconsistency regarding heterogeneity across the studies (Table 3). A sensitivity analysis showed that the omission of any of the included studies did not significantly affect the results.

### 3.5. Meta-Analysis

A meta-analysis was used to analyse if the prevalence of erosive tooth wear was associated with the gender of the patient, the presence or absence of digestive disorders, birthplace, caries prevalence, level of parental education, and frequency of toothbrushing.

#### 3.5.1. Gender

Eight studies contributed to this meta-analysis [15,24,29,33,35,36,40,41]. The results for the meta-analysis indicate that the likelihood of boys having erosive tooth wear is significantly higher than that of girls (OR = 1.26; 95% CI = 1.13, 1.41; *p* < 0.001). The heterogeneity across the studies is low (I^2^ = 0.0%, *p* = 0.492) (Appendix A). The quality of evidence was low as it was downgraded once due to observational data (Table 3).

#### 3.5.2. Digestive Disorders

Four studies [32,33,36,40] contributed to this meta-analysis. The likelihood of children with GERD, frequent vomiting, and/or digestive disorders having erosive tooth wear is significantly higher than that of children without any digestive disorders (OR = 1.38; 95% CI = 1.12, 1.70; *p* = 0.002) (Appendix A). Despite minimal heterogeneity across the studies (I^2^ = 0.0%, *p* = 0.413), the quality of evidence according to the GRADE assessment was considered low due to observational data (Table 3).

#### 3.5.3. Birthplace

Three studies were included in this meta-analysis [15,29,40]. No statistically significant difference was found in the likelihood of having erosive tooth wear between locals and foreigners (OR = 1.36; 95% CI = 0.46, 3.98; *p* = 0.579; I^2^ = 94.2%, *p* < 0.001) (Appendix A). Due to considerable heterogeneity across included studies and observational data, the quality of evidence was rated as very low (Table 3).

#### 3.5.4. Caries Prevalence

Three studies [15,33,36] were included in the meta-analysis. No statistical difference was found in the likelihood of having erosive tooth wear between children with caries and children without caries (OR = 0.97; 95% CI = 0.60, 1.56, *p* = 0.886) (Appendix A). There was considerable heterogeneity across included studies (I^2^ = 82.2%, *p* = 0.004). Following the GRADE recommendation, the body of evidence was very low due to observational data and substantial inconsistency between the included studies (Table 3).

#### 3.5.5. Parental Education

Three studies [15,27,40] were included in the meta-analysis comparing erosive tooth wear in children of parents with primary school level education and below compared with children of parents with education above primary level. There was no significant difference in likelihood of erosive tooth wear between children of parents with primary education and below, and those with education above primary level (OR = 0.98, 95% CI = 0.51, 1.90, *p* = 0.962; I^2^ = 82.5%, *p* = 0.003) (Appendix A).

Three studies [7,33,40] were included in the meta-analysis comparing erosive tooth wear in children of parents with secondary school level education and below compared with children of parents with education above secondary level. There was also no significant difference in the likelihood of erosive tooth wear between children of parents with secondary education or below and children of parents with education above secondary level (OR = 0.84, 95% CI = 0.51, 1.38, *p* = 0.498) (Appendix A). The level of heterogeneity across studies for both factors were noted to be considerable (I^2^ = 91.9%, *p* < 0.001). Due to the use of observational data and inconsistency as seen from I^2^ statistics and χ^2^ test on heterogenicity, the quality of evidence was rated as very low for both factors of erosive tooth wear (Table 3).

#### 3.5.6. Toothbrushing

Three studies [15,33,40] were included in the meta-analysis that showed there was no significant difference in the likelihood of erosive tooth wear in children who brush their teeth once or less than once a day compared with children who brush their teeth more than once a day (OR = 0.88, 95% CI = 0.73, 1.06, *p* = 0.168; I^2^ = 0.0%, *p* = 0.457) (Appendix A). According to the GRADE assessment, the quality of evidence was low as the data was observational in nature (Table 3).

### 3.6. Meta-Regression

Meta-regression was used to determine if the prevalence of erosive tooth wear was associated with confounders including age, different epidemiological erosion indices, sample size, human developmental index (HDI), and year of recruitment (Table 4).

#### 3.6.1. Age

Fifteen studies [13,15,26,28,29,31,32,33,34,35,36,37,39,40,41] were included in the meta-regression to examine any relationship between age and prevalence of erosive tooth wear. There was no significant difference in the prevalence of erosive tooth wear in different age groups when compared with children 6 years of age (Table 4).

#### 3.6.2. Erosion Indices

All twenty-two studies were included in the analysis of the association between erosion index used and prevalence of erosive tooth wear. There was no significant difference in the prevalence of erosive tooth wear when different erosion indices were used (Table 4).

#### 3.6.3. Sample Size

All twenty-two studies were included in the analysis of relationship between sample size of study and measured prevalence of erosive tooth wear. There was no significant difference in the prevalence of erosive tooth wear among studies with different sample sizes (Table 4).

#### 3.6.4. Human Development Index (HDI)

All twenty-two studies were included in the analysis to explore any correlations between HDI of the country where the study was conducted and the respective prevalence of erosive tooth wear. There was no significant difference in the prevalence of erosive tooth wear among studies conducted in countries of different human development index (HDI) tiers (Table 4).

#### 3.6.5. Year of Recruitment

All twenty-two studies were included in the analysis regarding relationship between year of recruitment of subjects and the prevalence of erosive tooth wear. There was no significant difference in the prevalence of erosive tooth wear among studies with different periods of recruitment of subjects for the studies (Table 4).

### 3.7. Narrative Review

A narrative review was conducted to study whether frequency of fruit juice and soft drinks intake affected the risk of erosive tooth wear.

#### 3.7.1. Fruit Juice Frequency

Seven studies [13,24,30,31,33,36,39] examined the association between the frequency of intake of fruit juice and the presence of erosive tooth wear.

Two studies [36,39] found that there was no significant difference in the prevalence of erosive tooth wear between children with a higher frequency intake of fruit juice and children with a lower frequency intake of fruit juice. Murakami et al. (2011) found that there was no significant difference between children who did not consume fruit juice and those who consumed them once a day, twice a day, or more than 3 times a day (*p* = 0.083). Raza and Hashim (2012) found no association between erosive tooth wear and the frequency of consumption of acidic drinks, which included fruit juice and other fruit-based drinks.

Five studies [13,24,30,31,33] found that a higher frequency of intake of fruit juice was significantly associated with a higher prevalence of erosive tooth wear in children. Three studies [24,30,31] found that children who consumed acidic fruit juice or fruit squash daily or more had significantly higher rates of erosive tooth wear. Maharani et al. (2019) and Nayak et al. (2011) also found that children with higher frequencies of intake of fruit juice had a higher incidence of erosive tooth wear (*p* = 0.001, *p* < 0.001, respectively). Another study [41] found that, among the children with erosive tooth wear, those who had a higher frequency of intake of fruit juices were associated (*p* = 0.048) with a greater severity of erosive tooth wear.

#### 3.7.2. Soft Drinks Frequency

Four studies [27,31,36,39] included in this systematic review analysed the relationship between the frequency of intake of soft drinks and the prevalence of erosive tooth wear. Only one study [39] found no association between the frequency of consumption of carbonated drinks and the prevalence of erosive tooth wear. The other three studies [27,31,36] found that the higher the frequency of intake of soft drinks, the more likely the child would have erosive tooth wear.

Children who had carbonated drinks once a day or more had a significantly higher prevalence of erosive tooth wear compared with children who had a lower frequency of carbonated drinks [27,31]. Another study by Murakami et al. (2011) found that children who consumed soft drinks twice a day and more than 3 times a day had 1.73 and 1.82 times higher likelihood of having erosive tooth wear compared with children who do not consume soft drinks. This finding was statistically significant (*p* = 0.023). Tschammler et al. (2016) found that the severity of erosive tooth wear was associated with a higher frequency of intake of lemonade or coke (*p* = 0.043).

## 4. Discussion

The combined result of this meta-analysis showed that the overall estimated prevalence of erosive tooth wear in children below 7 years old is 39.64%. This estimate is higher than those from other recently published reviews on erosive tooth wear [42,43]. A meta-analysis on primary teeth [42] showed a lower range of prevalence of erosive tooth wear from 5% to 35%. However, Corica and Caprioglio (2014) only included three studies, resulting in a much smaller sample of children included in the meta-analysis. Besides that, the erosion indices utilized in the included studies only examined maxillary anterior incisors without including the posterior teeth, which might account for the lower prevalence of erosive tooth wear in the study [42]. Furthermore, the study did not perform any meta-regression and subgroup analyses to evaluate potential confounders associated with erosion. Salas et al. (2015) also identified a lower estimated prevalence of erosive tooth wear. However, the study [42,43] was conducted on permanent dentition of children and adolescents aged 8 to 19 years old. Primary teeth are reported to be less resistant to erosive tooth wear than permanent teeth [3,44], which might have explained the higher overall estimated prevalence of erosive tooth wear in this study compared with the review by Salas et al. (2015).

Interestingly, this study found that the likelihood of boys having erosive tooth wear is significantly higher than girls (OR = 1.26; 95% CI = 1.12, 1.40; *p* < 0.001). Similar findings were also reported among adolescents. In adolescents, it is hypothesized that, since males tend to have higher physical activity [45], they are more prone to salivary changes [46]. Decreased salivary flow during exercise, as well as lower stimulated salivary flow rate, cause a decrease in clearance rate, leading to an increase in risk of erosive tooth wear [47]. Similarly, among preschool children, boys are found to engage in physical activity more often than girls [48]. Similar salivary changes may occur in preschool boys, resulting in a higher likelihood of erosive tooth wear compared with girls. Future research can focus on any resultant changes of salivary composition after physical activity in children.

The likelihood of children with GERD, frequent vomiting, and/or digestive disorders having erosive tooth wear is significantly higher than that of children without such digestive disorders (OR = 1.38; 95% CI = 1.12, 1.70; *p* = 0.002). This shows the need for the dentist to carefully take a detailed medical history for each patient due to the close relationship between the patient’s medical and dental health. Likewise, children with erosive tooth wear should be screened for any GERD or digestive issues. If these children are screened positive, then an appropriate referral should be made to a gastroenterologist for further evaluation and management.

Our findings suggest no statistically significant relationship between age and prevalence of erosive tooth wear. This appears to contradict the results in a systematic review published by Kreulen (2010), which identified a linear relationship between age and erosive tooth wear [49]. However, Kreulen (2010) only looked into dentinal erosion rather than both enamel and dentinal erosion as a whole, which might explain the difference in the results. This review also found that children with a higher frequency of consumption of soft drinks and fruit juices had higher prevalence of erosive tooth wear. While it is concerning that many children have poor dietary habits that lead to erosive tooth wear, raising awareness and educating parents will encourage children to reduce these erosive beverages. For instance, one-to-one dietary interventions given by medical and dental professionals [50] were found to be effective in motivating patients to make dietary changes. Parents should be educated during dental appointments on which foods can cause dental caries and/or erosive tooth wear.

Erosive medications that cannot be avoided may be a more worrying cause of erosive tooth wear. Oral medications for children are often in liquid form for ease of swallowing. They are also high in sugar to increase palatability, acceptability, and compliance of paediatric patients. Unfortunately, many of these medications have high erosive potential [51]. These medications have been implicated in softening enamel and increasing risk of erosive tooth wear [52,53]. Unfortunately, this review did not find sufficient studies on the association between intake of medications and erosive tooth wear that met the inclusion criteria. Future research can focus on whether the frequent intake of certain paediatric oral liquid medications may increase the risk of erosive tooth wear.

Some of the included studies recruited study participants from hospitals and clinics. As such, these participants may have pre-existing medical and dental conditions that may be confounders for associated factors of erosive tooth wear. Furthermore, to include a larger number of subjects in this review, studies using different erosion indices were all included in this study. While this study showed that there was no statistically significant difference in erosive tooth wear when different indices were used, some other studies have concluded otherwise. For example, a 2010 systematic review [49] suggested that erosion indices that focused on incisors only could lead to an underestimation of prevalence of severe wear. Another systematic review and meta-regression analysis [43] found that TWI presents the highest prevalence rates of erosive tooth wear. Moreover, the use of different indices precluded an analysis on the severity of erosive tooth wear, as different indices had different standards of severity. As the use of different erosion indices could result in variability in the estimates of prevalence of erosive tooth wear, there is a real need for an agreement among researchers on which index to use. Such standardized research methods will ensure that results are less heterogeneous and thus allow for easier analysis and comparison between studies.

Medical and dental professionals should strive to help raise awareness of erosive tooth wear as well as to educate the public on good oral and dietary habits. Healthcare professionals who see patients who have GERD or digestive issues should be aware of the higher risk of having erosive tooth wear in these children and provide appropriate counselling and advice. Other than cases of erosive tooth wear caused by GERD, digestive issues, and medications, increasing evidence has suggested that underlying developmental dental defects might also increase the risk of erosive tooth wear among children [54]. As these causes of erosive tooth wear may not be so easily controlled, future research should also look into different preventive measures, such as varnishes that can be applied to patients to prevent erosive tooth wear. The role of fluoride in preventing erosive tooth wear and in encouraging remineralization is well-researched and understood. However, there is no concrete evidence yet on whether fluoride varnishes can prevent erosive tooth wear. Therefore, future studies should look into the possibility of fluoride varnishes and other varnishes such as arginine-enhanced fluoride, and CPP-ACP in preventing or reducing erosive tooth wear caused by acids that may be intrinsic, or even into other acidic liquids such as paediatric oral liquid medications, acidic fruit juices, and soft drinks.

The strengths of this systematic review included having two independent reviewers in screening papers for inclusion with almost ideal agreement (κ = 0.937). A risk of bias assessment with JBI critical appraisal checklist, a heterogeneity assessment, sensitivity analyses, and publication bias analyses were also conducted, with the GRADE approach being used to evaluate the quality of evidence for each outcome. However, this study also has its limitations. Published studies without full texts in English or studies in which full texts cannot be retrieved despite attempts to contacting the respective corresponding authors had to be excluded. This might possibly lead to inclusion bias, especially for studies from non-English-speaking countries.

## Figures and Tables

**Figure 1 healthcare-10-00491-f001:**
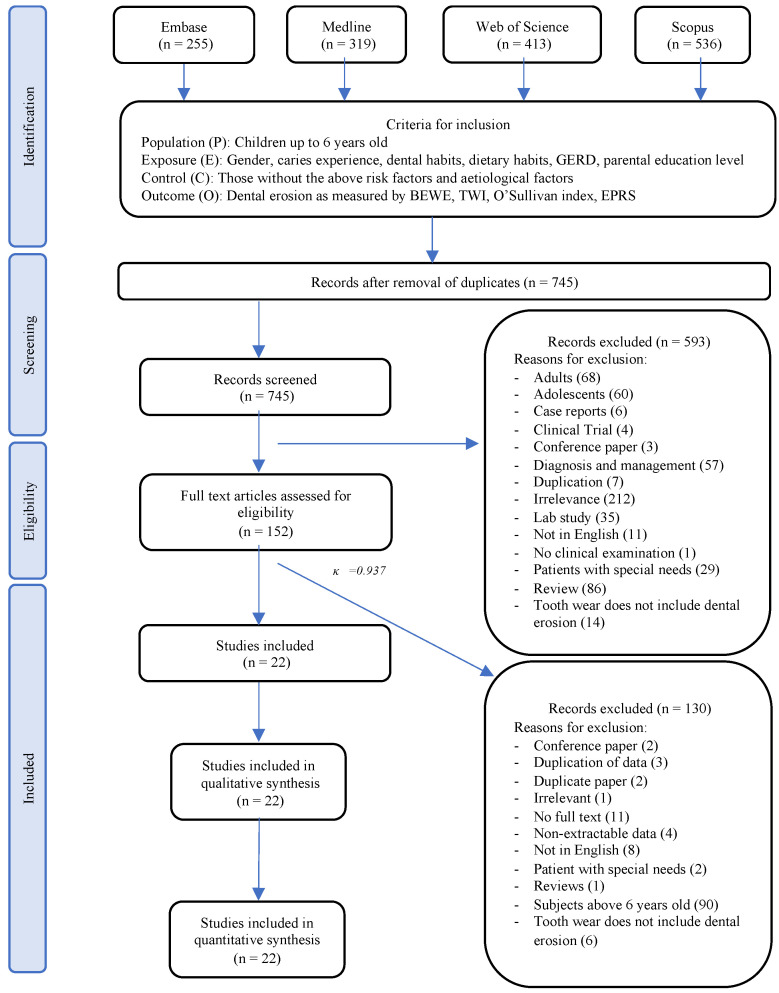
PRISMA flowchart of the screening and study selection process.

**Figure 2 healthcare-10-00491-f002:**
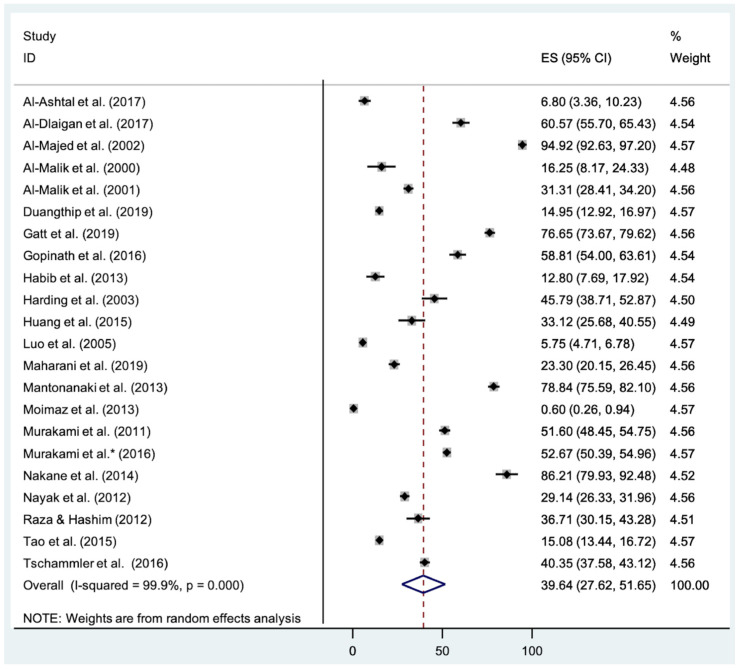
Meta-analysis and forest plot showing combined prevalence rates of all included studies. 95% CI: 95% Confidence Interval; ES: Effect size; p: *p*-value.

**Table 1 healthcare-10-00491-t001:** Characteristics of included studies.

No.	Author (Published Year, Country Where Study Was Conducted)	Study Design	Number of Subjects (% Males)	Age Range (Year)	Recruitment	Inclusion (I)/ Exclusion (E) Criteria	Factors Evaluated	Erosion Index Used
1	Al-Ashtal (2017, YE)	Cross-Sectional	206 (NR)	5–6	University Dental Clinic	NR	nil	EPRS
2	Al-Dlaigan (2017, SB)	Cross-Sectional	388 (47)	3–5	Kindergartens (2 public, 8 private)	NR	Diet	TWI
3	Al-Majed (2002, SB)	Cross-Sectional	354 (100)	5–6	Elementary Schools	(E) Children without questionnaires (E) Children who were not examined	Diet OH	TWI
4	Al-Malik (2000, SB)	Cross-Sectional	80 (64)	4–5	Home for Disadvantaged	(E) Children with significant medical history/learning difficulties (none)	Caries	TWI
5	Al-Malik (2001, SB)	Cross-Sectional	987 (NR)	2–5	Schools (6 public, 11 private)	(E) Children without consent forms (E) Absent for examination (E) Uncooperative for examination	Diet OH Social Class	TWI
6	Duangthip (2019, HK)	Cross-Sectional	1204 (46)	3–5	7 non-profit kindergartens	(I) Healthy children (E) Children with chronic diseases (E) Children with special needs (E) Below 3 years old (E) Uncooperative for examination (E) Absent for examination	Gender Age SES Parental education level Diet OH Caries	BEWE
7	Gatt (2019, MT)	Cross-Sectional	775	3–5	Schools (state, church, independent)	(I) Resident on Islands all their lives 3–5 years old (E) Children with enamel defects exhibiting loss of tooth tissue	Gender Age SES Parental education level Diet OH Caries Asthma/respiratory disease GERD Medications BMI	BEWE
8	Gopinath (2016, AE)	Cross-Sectional	403 (48.14)	5	Kindergartens	NR	Gender Diet Caries Nationality	TWI
9	Habib (2013, US)	Cross-Sectional	164	2–4	Daycare centre Preschool Grade school	(I) Consent given (I) Resident of Kansas City Metropolitan area	Gender EthnicitySES OH Diet	TWI
10	Harding (2003, IE)	Cross-Sectional	202	5	Schools	(E) Medical condition (E) Children on long term oral or inhaled corticosteroids	Gender SES Diet OH Fluoridation Vomiting	TWI
11	Huang (2015, AU)	Cohort	154 (45)	2–4	Public birthing and community health clinics	(E) Those who did not attend all 3 reviews	Social Medical history Dental and dietary habits GERD	TWI
12	Luo (2005, CN)	Cross-Sectional	1949	3–5	Kindergartens	(I) No gastrointestinal problems	SES Parental Education Diet	TWI
13	Maharani (2019, ID)	Cross-Sectional	691 (53.54)	5	Kindergartens	(E) Uncooperative for examination (E) No questionnaire (E) No consent	Gender SES Parental education SES Diet OH Digestive disorders	BEWE
14	Mantonanaki (2013, GR)	Cross-Sectional	524 (examination and questionnaire done)605 (examination only)	5	Kindergartens	(E) No examination (E) Incomplete questionnaire (E) Immigrants (E) Above 5 years old	Parental education level SES OH Vomiting/regurgitation/heartburn Medication	BEWE
15	Moimaz (2013, BR)	Cross-Sectional	1993 (49.42)	4–6	Preschools (public)	(E) No consent (E) Uncooperative for examination (E) Absent for examination	Gender Age OH	TWI
16	Murakami (2011, BR)	Cross-Sectional	967 (47.88)	3–4	Children attending a statutory National Children’s Vaccination day	(E) Children living in same household as selected child (E) Children without parents present	Age Caries SES Diet Acid reflux	TWI
17	Murakami (2016, BR) (some repeat data from 2011, repeat data excluded in statistical analysis)	Cross-Sectional	2801	3–4	Children attending a statutory National Children’s Vaccination day in 2008, 2010, 2012	(E) Children living in same household as selected child (E) Children without parents present	nil	TWI
18	Nakane (2014, JP)	Cross-Sectional	116 (57.76)	2–6	University Hospital Paediatric Dental Clinic	NR	SES Diet Vomiting Medication OH Fluoride Medical history	O’Sullivan Index
19	Nayak (2012, IN)	Cross-Sectional	1002	5	Schools	(E) Special health care needs(E) Orofacial defects	Diet OH	SES Diet
20	Raza & Hashim (2012, AE)	Cross-Sectional	207 (46.4)	5–6	Schools (private)	(I) Children who completed examination and questionnaire	Age Ethnicity Mother education level Diet Medications GERD Swimming	TWI
21	Tao (2015, CN)	Cross-Sectional	1837 (51.55)	3–6	Kindergartens	(E) Children with orthodontics appliances	Age Gender Parental education Diet OH Medical health SES Swimming	O’Sullivan Index
22	Tschammler (2016, DE)	Cross-Sectional	775 (52.26)	3–6	Kindergartens	(E) No consent Uncooperative during examination	Diet Chronic illness OH	BEWE

Key: NR: Not reported. OH: Oral hygiene. SES: Socioeconomic status. EPRS: Erosion partial recording system. TWI: Tooth wear index. BEWE: Basic Erosive Wear Examination. Country Alpha-2 Codes. AE: United Arab Emirates. AU: Australia. BR: Brazil. CN: China. DE: Germany. GR: Greece. HK: Hong Kong. ID: Indonesia. IE: Ireland. IN: India. JP: Japan. MT: Malta. SB: Solomon Islands. US: United States of America. YE: Yemen.

**Table 2 healthcare-10-00491-t002:** Risk of bias of included studies.

No.	Author	Year	Qs 1	Qs 2	Qs 3	Qs 4	Qs 5	Qs 6	Qs 7	Qs 8	Qs 9	Total Score	Risk of Bias
1	Al-Ashtal et al.	2017	2	1	1	2	3	1	1	1	3	5 out of 9	Moderate
2	Al-Dlaigan et al.	2017	1	1	1	2	2	1	1	1	3	6 out of 9	Moderate
3	Al-Majed et al.	2002	1	3	1	2	1	1	1	1	1	7 out of 9	Low
4	Al-Malik et al.	2000	1	1	1	1	1	1	1	1	0	9 out of 9	Low
5	Al-Malik et al.	2001	1	1	1	1	1	1	1	1	3	8 out of 9	Low
6	Duangthip et al.	2019	1	1	1	1	1	1	1	1	1	9 out of 9	Low
7	Gatt et al.	2019	1	1	1	2	1	1	1	1	2	7 out of 9	Low
8	Gopinath et al.	2016	1	2	3	2	1	1	1	1	1	6 out of 9	Moderate
9	Habib et al.	2013	1	1	1	1	3	1	1	1	3	7 out of 9	Low
10	Harding et al.	2003	1	3	3	1	1	1	1	1	3	6 out of 9	Moderate
11	Huang et al.	2015	1	1	3	2	3	1	1	1	3	5 out of 9	Moderate
12	Luo et al.	2005	1	1	3	1	3	1	1	1	3	6 out of 9	Moderate
13	Maharani et al.	2019	1	1	1	1	3	1	1	1	1	8 out of 9	Low
14	Mantonanaki et al.	2013	1	1	1	1	3	1	1	1	3	7 out of 9	Low
15	Moimaz et al.	2013	1	3	3	2	2	1	1	1	3	4 out of 9	High
16	Murakami et al.	2011	1	1	1	1	1	1	1	1	1	9 out of 9	Low
17	Murakami et al.	2016	1	1	1	2	3	1	1	1	1	7 out of 9	Low
18	Nakane et al.	2014	2	1	3	1	3	1	1	1	3	5 out of 9	Moderate
19	Nayak et al.	2012	1	1	1	1	3	1	1	1	1	8 out of 9	Low
20	Raza & Hashim	2012	1	1	3	1	3	1	3	1	3	5 out of 9	Moderate
21	Tao et al.	2015	1	1	1	1	1	1	1	1	3	8 out of 9	Low
22	Tschammler et al.	2016	1	1	1	2	1	1	1	1	1	8 out of 9	Low

Key: Qs 1: Question 1—Was the sample frame appropriate for addressing the target population? Qs 2: Question 2—Were study participants sampled in an appropriate way? Qs 3: Question 3—Was the sample size adequate? Qs 4: Question 4—Were the study subjects and the setting described in detail? Qs 5: Question 5—Was the data analysis conducted with sufficient coverage of the identified sample? Qs 6: Question 6—Were valid methods used for the identification of the condition? Qs 7: Question 7—Was the condition measured in a standard, reliable way for all participants? Qs 8: Question 8—Was there appropriate statistical analysis? Qs 9: Question 9—Was the response rate adequate, and if not, was the low response rate managed appropriately? 1—Yes, 2—No, 3—Unclear, and N/A—Not applicable.

**Table 3 healthcare-10-00491-t003:** GRADE summary of findings.

Outcome	No. of Studies	No. of Participants	Results	Risk of Bias ^†^	Inconsistency ^‡^		Indirectness ^§^	Imprecision ^¶^	Publication Bias ^††^	Quality of Evidence (GRADE)
					I^2^ Statistics	Heterogenicity χ^2^ Test (*p* Value)				
Dental Erosion Prevalence	22	17,300	Estimated overall prevalence: 38.38% *p* < 0.001	Not serious	99.9% *	*p* < 0.001 ***	Not serious	Not serious	Not serious	⊕OOO very low due to observational data, substantial inconsistency
				–	↓		–	–	–	
Gender	8	1106	Likelihood of boys have dental erosion is significantly higher than girls (*p* < 0.001)	Not serious	0.0%	*p* = 0.492	Not serious	Not serious	N/A	⊕⊕OO low due to observational data
				–	–		–	–		
GERD	4	227	Likelihood of children with GERD/frequent vomiting/digestive disorders having dental erosion is higher than children without the above disorders (*p* = 0.002)	Not serious	0.0%	*p* = 0.413	Not serious	Not serious	N/A	⊕⊕OO low due to observational data
				–	–		–	–		
Birthplace	3	243	No significant difference	Not serious	94.2% *	*p* < 0.001 ***	Not serious	Not serious	N/A	⊕OOO very low due to observational data, substantial inconsistency
				–	↓		–	–		
Dmft > 0 /Caries Experience	3	346	No significant difference	Not serious	I^2^ = 82.2% *	*p* = 0.004 **	Not serious	Not serious	N/A	⊕OOO very low due to observational data, substantial inconsistency
				–	↓		–	–		
Parental Education (primary)	3	114	No significant difference	Not serious	I^2^ = 82.5% *	*p* = 0.003 **	Not serious	Not serious	N/A	⊕OOO very low due to observational data, substantial inconsistency
				–	↓		–	–		
Parental Education (Secondary)	3	442	No significant difference	Not serious	I^2^ = 91.9% *	*p* < 0.001 ***	Not serious	Not serious	N/A	⊕OOO very low due to observational data, substantial inconsistency
				–	↓		–	–		
Toothbrushing	3	231	No significant difference	Not serious	I^2^ = 0.0%	*p* = 0.457	Not serious	No serious	N/A	⊕⊕OO low due to observational data
				–	–		–	–		
				–	–		–	–		

* *p* < 0.05, ** *p* < 0.01, *** *p* < 0.001. GRADE: Grading of Recommendations Assessment Development and Evaluation; ↓: Downgrade by one level in quality of evidence; –: No change in quality of evidence. ^†^ Risk of bias: If half or more of the studies have serious risk, then the overall risk of bias is considered serious. ^‡^ Inconsistency: If I^2^ ≥ 70% (*), and *p*-value of χ^2^ test < 0.05 (**), then the overall inconsistency is considered serious. ^§^ Indirectness: If the applicability of findings was limited due to population, intervention, comparator, and outcomes, then the overall indirectness is considered serious. ^¶^ Imprecision: If the total number of events for dichotomous outcomes < 300, and the total number of events for continuous outcomes < 400, then the overall imprecision is considered serious. ^††^ Publications bias: If the *p*-value of Begg’s funnel plot < 0.05, then the overall publication bias is considered to be serious. If the funnel plot could not be constructed due to the limited numbers of studies included, then it was considered not applicable (N/A).

**Table 4 healthcare-10-00491-t004:** Association between factors (age, indices, sample size, HDI, and year of recruitment) and prevalence of erosive tooth wear.

Variables	N (Studies)	Prevalence (%)	Meta-Regression
**Age**			
3	6	34.4	0.900
4	8	30.4	0.859
5	12	38.9	0.635
6	4	32.7	Reference
**Indices**			
BEWE	5	46.81	0.197
TWI	14	37.87	0.289
O’Sullivan	2	50.58	0.208
EPRS	1	6.80	Reference
**Sample Size**			
1–499	10	45.22	0.106
500–999	5	52.34	0.074
1000–1499	3	28.12	0.640
1500+	4	18.48	Reference
**Human Development Index (HDI)**			
Below 0.55	1	6.8	0.151
0.55–0.69	5	46.50	0.899
0.7–0.79	6	24.78	0.101
0.8–1.0	10	48.43	Reference
**Year of Recruitment**			
Before 2010	7	40.88	0.748
2010–2014	8	43.66	0.601
2015 and later	8	36.12	Reference

## Data Availability

All data generated or analysed during this study are included in this article and its Appendix A files. Further enquiries can be directed to the corresponding author.

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
