# Peer review of "Prevalence and Associated Factors of Erosive Tooth Wear among Preschool Children—A Systematic Review and Meta-Analysis"

_healthcare, 2022, doi:10.3390/healthcare10030491_

Round 1

Reviewer 1 Report

1. The subject of your research is preschool children up to 6 years. Please explain why you included 'adolescent', 'teenage' in your search terms.

2. Twenty-two articles were included in the analysis, but not all papers were included in table1. Please include all studies.

3. Please add a description of the index used in this study as a measurement index of dental erosion.

Author Response

We are very grateful for all the suggestions made by Reviewer 1.  Point by point, We have addressed Reviewer 1's suggestions in the attached file.

Reviewer 2 Report

The aim of this study was to systematically review the literature to identify the prevalence and associated factors of erosive tooth wear among children. However, the article needs to make some changes before final acceptance.

  1. Some terms should be replaced eg gender for sex.
  2. Replace your keywords for Mesh terms whenever possible for example, replace erosion for “Tooth Erosion”
  3. Clarify whether you included studies with 6 or 7 years. Discrepant information on lines 76 and 81.
  4. Search phrases must be presented individually by each database in order to allow replication of the search
  5. Authors should not limit their search to July 2020 since at least four articles in the literature have been published after that and they had eligibility to be included.
  6. Did the authors try to contact the corresponding authors of articles where the full text was not available? This information must be clarified.
  7. In figure 1, the new flowchart of systematic reviews should be used, where the grey literature is also mentioned.
  8. Table 1 included only 6 articles, but this systematic review included 22 articles as a final sample. The table must be reformulated and must include all included studies. Consider adding a new column with main conclusions.
  9. References must be reformulated according to the journal's rules.

Author Response

We are very grateful for all the constructive comments given by Reviewer 2. Point by point, we have addressed each comment accordingly with the “track-changes” function, as requested by the jounral. We apologize due to the “track changes “function, there might be some discrepancies in the line number mentioned below with the main text file.

Reviewer 3 Report

Dear authors,

this is an interesting manuscript, talking about Prevalence and Associated Factors of Erosive Tooth Wear among Preschool Children. The review was well-conducted and the methods are well structured. However, I suggested the authors to do some minor corrections and changes in order to make this work more clear and readable:

  • Abstract should be reorganised: As journal guidelines, the abstract should be a single paragraph and should follow the style of structured abstracts, but without headings. Please change the abstract
  • The full manuscript should be checked for minor errors. 

    Line 29: Take off the reference “[2]”, it’s a repetition

    Line 35-37: “Erosive tooth… tooth wear [5].” Please correct

    Line 62: After “… CRD42020186982.” close parenthesis. 

    Line 76-84: it would be better written by listing “Inclusion criteria: …” and “Exclusion criteria: …”. 

    Line 92: There is a repetition of the word “pediatric”. 

    Line 104: There is a repetition “(first and second authors)”, already declared at line 97. 

    Line 116-118: There is a repetition “Kappa (k)… reviewers.”, already declared at line 100-102. Please adjust

    Line 148: Add full-stop. 

    Line 150: There are numbers organization, while for Materials and Methods are absent (lines 60, 65, 85, 95, 103, 109, 119, 132, 137, 140). 

    Line 175: there is a second full-stop. 

    Between line 175-176: Table 1, add what “NR” abbreviation stands for. 

    Between line 208-209: Table 2, correct word “appropriate”, correct word “ina”, at line 113 there is also “not applicable”. 

    Line 222: there is a second full-stop. 

    Line 383: Publish by… Please add the author, not only the reference

    Line 434: and a high Kappa coefficient was achieved. Please correct

  • In your study there is no evaluation of the developmental dental defects associated with tooth wear or a citation of that. Please read and also use the article "Grande, F., & Catapano, S. (2021). Developmental Dental Defects and Tooth Wear: Pathological Processes Relationship. In Human Teeth–Structure and Composition of Dental Hard Tissues and Developmental Dental Defects. IntechOpen."

Author Response

We would like to thank Reviewer 3 for all his/her detailed suggestions and comments.

Point by point, we have addressed each comment accordingly with the “track-changes” function, as requested by the jounral. We apologize due to the “track changes “function, there might be some discrepancies in the line number mentioned below with the main text file.
